# Prognostic Role of the Removed Vaginal Cuff and Its Correlation with L1CAM in Low-Risk Endometrial Adenocarcinoma

**DOI:** 10.3390/cancers14010034

**Published:** 2021-12-22

**Authors:** Enrico Vizza, Valentina Bruno, Giuseppe Cutillo, Emanuela Mancini, Isabella Sperduti, Lodovico Patrizi, Camilla Certelli, Ashanti Zampa, Andrea Giannini, Giacomo Corrado

**Affiliations:** 1Gynecologic Oncology Unit, Department of Experimental Clinical Oncology, IRCCS “Regina Elena” National Cancer Institute, 00144 Rome, Italy; valentina.bruno@ifo.gov.it (V.B.); giuseppe.cutillo@ifo.gov.it (G.C.); emanuela.mancini@ifo.gov.it (E.M.); camilla.certelli@hotmail.it (C.C.); ashanti.zampa@ifo.gov.it (A.Z.); 2Scientific Direction, IRCCS “Regina Elena” National Cancer Institute, 00144 Rome, Italy; isabella.sperduti@ifo.gov.it; 3Department of Surgery, Section of Gynaecology and Obstetrics, Tor Vergata University, 00133 Rome, Italy; lodovicopatrizi@gmail.com; 4Department of Medical and Surgical Sciences and Translational Medicine, “Sapienza University”, Rome 00185, Italy; andrea.giannini@uniroma1.it; 5Dipartimento Scienze della Salute della Donna, del Bambino, e di Sanità Pubblica, Ginecologia Oncologica, Fondazione Policlinico Universitario A. Gemelli IRCCS, 00168 Rome, Italy; giacomo.corrado@policlinicogemelli.it

**Keywords:** low-risk endometrial cancer, vaginal cuff, L1CAM

## Abstract

**Simple Summary:**

In assessing the risk factors for the recurrence of EC, the length of the vaginal cuff removed during surgery has shown discrepant results. The aim of this study was to investigate the role of the excised vaginal cuff length as a prognostic factor and its correlation with the expression of L1CAM. To our knowledge, this is the first study evaluating this prognostic factor in such a secluded and tidy cohort. According to our results, vaginal cuff length does not seem to be an independent variable in an EC low-risk group in terms of DFS. Moreover, L1CAM seems to be associated with a higher risk of distant recurrence. A prospective randomized trial in an EC low-risk group is needed to confirm its negative prognostic role, and to determine its potential value in clinical practice to detect that subgroup of low-risk patients which is at a higher risk of recurrence, especially at distant.

**Abstract:**

Objective: The aim of our study was to investigate the role of the excised vaginal cuff length as a prognostic factor in terms of DFS and recurrence rate/site, in low-risk endometrial cancer (EC) patients. Moreover, we correlated the recurrence with the expression of L1CAM. Material and Methods: From March 2001 to November 2016, a retrospective data collection was conducted of women undergoing surgical treatment for low-risk EC according to ESMO-ESGO-ESTRO consensus guidelines. Patients were divided into three groups according to their vaginal cuff length: V0 without vaginal cuff, V1 with a vaginal cuff shorter than 1.5 cm and V2 with a vaginal cuff longer than or equal to 1.5 cm. Results: 344 patients were included in the study: 100 in the V0 group, 179 in the V1 group and 65 in the V2 group. The total recurrence rate was 6.1%: the number of patients with recurrence was 8 (8%), 10 (5.6%) and 3 (4.6%), in the V0, V1 and V2 group, respectively. No statistically significant difference was found in the recurrence rate among the three groups. Although the DFS was higher in the V2 group, the result was not significant. L1CAM was positive in 71.4% of recurrences and in 82% of the distant recurrences. Conclusions: The rate of recurrence in patients with EC at low risk of recurrence does not decrease as the length of the vaginal cuff removed increases. Furthermore, the size of the removed vaginal cuff does not affect either the site of recurrence or the likelihood of survival.

## 1. Introduction

Endometrial cancer (EC) is the commonest gynecological cancer in postmenopausal women in developed countries [1]. Well-established clinical pathological risk factors, utilized in the present treatment guidelines, are age, histological type, tumor grade, International Federation of Gynecology and Obstetrics (FIGO) stage [2], depth of myometrial invasion, and lymph-vascular space invasion (LVSI). Consistent with clinical pathological prognostic factors described within the ESMO-ESGO-ESTRO consensus guidelines, patients were divided into four classes: low, intermediate, intermediate high and high risk of recurrence [3].

Nevertheless, these parameters are not sufficient to predict either the outcome or the recurrence rate for patients in FIGO stage I, with only 5–20% of the patients treated with a total hysterectomy and bilateral salpingo-oophorectomy showing vaginal and pelvic recurrence [4,5]. Among other poorly investigated risk factors, the length of the vaginal cuff removed during surgery has shown discrepant results. Women with low or low-intermediate risk EC are treated with surgery alone [6], consisting of type A radical hysterectomy and bilateral salpingo-oophorectomy with less than 10 mm of vaginal resection [7].

Recently, it has been shown that specific genetic markers [8] (L1 cell adhesion molecule (L1CAM), Anexin 2, insulin-like growth factor receptor, epidermal growth factor receptor, etc.) and aberrant molecular signaling pathways could be key players in cancer cells’ metastatic processes, although further clinical trials are needed to confirm their prognostic value in clinical practice [9]. Among these genetic markers, in a previous study we focused on L1CAM, a very promising potential prognostic factor in EC, since its expression has been already related to a poor DFS and OS in stage I endometrial cancer [10].

The aim of our study was to further investigate the role of the excised vaginal cuff length as a prognostic factor in terms of DFS and recurrence rate/site, in a homogenous group of only low-risk EC patients. Moreover, we correlated the recurrences with the expression of L1CAM.

## 2. Materials and Methods

We conducted a retrospective study on all women with endometrial cancer treated by the same surgical team between March 2001 and November 2016 at the Gynaecologic Oncologic Unit, IRCCS “Regina Elena” National Cancer Institute of Rome, Italy. Informed consent to oncological treatment and regarding research use of their medical information was obtained from all the patients in accordance with local and international legislation (Declaration of Helsinki) [11]. The research protocol was approved by the Ethics Committee of IRCCS “Regina Elena” National Cancer Institute, Rome, Italy (Project identification code: CE 470/12). All research data are openly available on proper request.

### 2.1. Study Design

The inclusion criteria were: adenocarcinoma endometrial cancer confirmed by definitive histological examination, patients classified as low risk according to ESMO-ESGO-ESTRO classification (FIGO stage IA, tumor grading 1 or 2, absence of lymphovascular space invasion), absence of adjuvant therapy after surgery, absence of pelvic or aortic lymphadenectomy during surgery and a minimum of 60 months of follow-up. The exclusion criteria were: patients who underwent surgery in centers different from our institution, patients who underwent adjuvant therapies after surgery, patients with a diagnosis of other types of cancers, patients with synchronous cancers and patients lost to follow-up.

Patients were divided in three groups according to the vaginal cuff length: V0 without vaginal cuff, V1 with a vaginal cuff shorter than 15 mm, and V2 with a vaginal cuff longer than or equal to 15 mm. These cut-offs were established in accordance with previous studies on this issue [3,7].

All patients were evaluated prior to surgery using medical records, physical examination, pelvic vaginal examination, chest X-rays, ultrasound scans and pelvic magnetic resonance imaging (MRI) scans. All the patients underwent a type A radical hysterectomy according to the Querleu-Morrow classification [6] and a bilateral salpingectomy with or without a bilateral oophorectomy according to their age and menopausal status. Data about age, body mass index (BMI), FIGO stage, recurrences, sites of recurrence and disease free survival (DFS) were collected. A gynecological pathologist examined and interpreted all surgical specimens. Low-risk EC was diagnosed after examination of permanent sections. The architectural classification was defined by the standard criteria of the International Federation of Gynaecology and Obstetrics (FIGO). Tumor size was macroscopically measured on fresh tissue by gynecological pathologists who noted the size in 3 major dimensions. The most important of the three tumor dimensions was defined as the primary tumor diameter [12]. LVSI has been defined for the presence of adenocarcinoma of any size in the canals lined by the endothelium of uterine samples extracted at the time of surgery [13]. Postoperative cancer surveillance included quarterly follow-up visits for the first 2 years and half-yearly visits thereafter. Biopsies were performed only in cases of suspicious findings and imaging studies in cases of suspected distant metastasis. If an isolated relapse was diagnosed, treatment with curative intent was initiated unless precluded by the patient or disease factors. When not available from medical records, follow-up data were collected through phone calls.

Recurrence was defined as documentation of metastasis by biopsy or imaging after a DFS ≥ 3 months. The primary relapse sites were grouped into 3 categories:Pelvic group: recurrence within pelvic area or vaginal cuff [14];Nodal group: recurrence in pelvic, para-aortic node and/or other node-bearing area [15];Distant group: disease recurring in the abdominal peritoneum and/or lung, liver, or other distant sites [16,17].

In cases of multiple concomitant localizations of relapse, the patient was allocated to the group with the most advanced disease. All women included in the study were followed up until death or until the start of the study period (March 3, 2021). Patient survival status was decided as alive with no evidence of disease (NED), alive with disease (AWD), death from inter current disease (DOID) and death from disease (DOD) at the time of the last follow-up. For all study subjects with a recorded death, this was confirmed by performing a search on the Social Security Death Rate.

### 2.2. Antibody and Immunohistochemistry

The mouse monoclonal antibody anti-L1CAM, clone UMAB48, was purchased from OriGene Technologies (Rockville, MD, USA). The formalin fixed paraffin-embedded (FFPE) tissue blocks were collected and 5 μm sections were dug and mounted on Superfrost slides. Antigen retrieval was performed at 96 °C (10 mM/L citrate buffer, pH 6) for 20 min. Sections were incubated with the first antibody anti-L1CAM (1:30) for a half-hour at the same temperature. Bond Polymer Refine Detection revealed an immunoreaction consistent with the manufacturer’s procedure (Leica Biosystems) in an automatic autostainer Bond III Leica Biosystems. Diaminobenzidine was used as a chromogenic substrate. A Nikon ECLIPSE 55i microscope with a HESP Technology camera was used. Scale bars were 50 µm. The expression level of L1CAM protein was analyzed by IHC analysis. Optimal cut-off was decided at a 20% positive staining for L1CAM [7].

### 2.3. Statistical Analysis

Descriptive statistics was used to describe the patients’ characteristics. Continuous and categorical variables were compared using One-Way Anova and chi-square tests or Fisher exact tests, as appropriate, respectively. All significance was defined at the *p* < 0.05 level. The SPSS (SPSS Inc., Chicago, IL, USA) and GraphPad Prism ver. 7.0 (GraphPad Software, San Diego, CA, USA) statistical programs were used for the analyses. DFS was calculated by the Kaplan–Meier product-limit method from the date of surgery until the time of relapse.

## 3. Results

During the study period, 1078 EC were treated at our National Cancer Centre. According to the inclusion criteria, 344 patients were included in the study: 100 in the V0 group, 179 in the V1 group and 65 in the V2 group. Patients’ clinical and pathological characteristics in the three groups are shown in Table 1. The median age was 58 (range, 30–85 years), 60 (range, 28–85 years) and 58 (range, 32–79 years), respectively, in the V0, V1 and V2 group, without significant differences in the three groups (*p* = 0.8). Furthermore, there were no significant differences in patients’ BMI (*p* = 0.17) with a median BMI of 28.9 (range 17–51), 30 (range 16–53), 28 (range 18–51), respectively, in the V0, V1 and V2 group. Furthermore, when the three groups were compared, the tumor grading was higher in V2 compared to V0 group (*p* < 0.01).

We identified 21 women who developed recurrent low-risk EC and the total recurrence rate was 6.1%. The median age was 68 (range, 50–75 years) and the median BMI was 27.4 kg/m^2^ (range, 20–40 kg/m^2^). Histologic grade was determined as grade 1 in 6 women (28.6%) whereas 15 patients had grade 2 histology (71.4%). Median primary tumor size was 20 mm (range, 15–34 mm). Six women (28.6%) were treated with an abdominal approach, fourteen (66.6%) with minimally invasive surgery and one with a vaginal surgery (4.8%). In our study, eight of the recurrences (38.1%) occurred within 3 years of primary surgery whereas the others occurred over 3 years after the initial diagnosis. The diagnosis of recurrence was biopsy-proven in 10 women (47.6%) whereas the others were only diagnosed by imaging studies. We observed 7 (33.3%) isolated vaginal recurrences, 3 (14.3%) nodal failures and 11 (52.4%) hematogenous disseminations. Overall, eight relapses (38.1%) were loco-regional while the others were extrapelvic. Treatment of the recurrences was applied according to the institutional practices at that time, and consisted of surgical resection plus radiotherapy (*n* = 7, 33.3%), surgical resection plus chemotherapy (*n* = 3, 14.3%) and chemotherapy (*n* = 11, 52.4%). The median DFS was 47 months (range, 5.4–130.7 months) and overall survival (OS) was 74.6 months (range, 10.9–141.2 months). At the time of reporting, 12 (57.2%) patients were dead of disease (DOD), 4 (19%) were alive with disease (AWD) and 5 (23.8%) were alive with no evidence of disease (NED). The median DFS was not statistically significant between loco-regional and extrapelvic recurrences (47 vs. 51.45 months; *p* = 0.3), as well as OS (60.7 vs. 75.25 months; *p* = 0.7). Moreover, OS was significantly higher for patients with DFS ≥ 36 months compared to those with DFS < 36 months (75.9 vs. 33.2 months, respectively; *p* = 0.01). Clinical characteristics of recurrences, median DFS and OS, type of salvage therapies and follow-up status are summarized in Table 2.

The number of patients with recurrence was 8 (8%), 10 (5.6%) and 3 (4.6%), in the V0, V1 and V2 group, respectively (Table 3 and Figure 1). No statistically significant difference was found in the recurrence rate in the three groups (Table 2 and Figure 1). The sites of recurrence in the three groups are shown in Table 3. Furthermore, due to the low numbers of recurrences per each subgroup a statistical comparison in terms of different sites of recurrence in the three groups was not applicable. However, in V0 group, there were 62.5% of distant recurrences, 25%, and 12.5% of, respectively, lymph nodes and pelvic recurrences. In V1 group, the 50% of recurrences were in pelvic site, 40% in distant site followed by lymph nodes (10%). In V2 group, there was only one pelvic and two distant recurrences. Although the DFS was higher in the V2 group, the result was not significant (*p* = 0.5) (Figure 2).

L1CAM was performed on all but two relapses and it was positive in 71.4% of recurrences. Furthermore, although a trend was only found in the statistical analysis (*p* = 0.078) probably due to the low numbers of recurrences, it was positive in 100% of the distant recurrences and in 57.1% of the pelvic recurrences (Figure 3).

## 4. Discussion

Our results showed that the rate of recurrence in patients with endometrial adenocarcinoma at low risk of recurrence and who have not undergone adjuvant therapy, does not decrease as the vaginal cuff length removed increases. Furthermore, the size of the removed vaginal cuff does not affect either the site of recurrence or survival.

The ILIADE randomized study has shown how in stage I EC the radicality of surgery, Piver–Rutledge class I versus Piver–Rutledge class II, did not affect the loco-regional recurrence rate, despite the longer length of the vaginal cuff removed in the class II arm than class I, in which the median length of the excised vaginal cuff was 15 mm [4]. Conversely, a retrospective study has demonstrated a correlation between DFS, OS and removed vaginal cuff, showing that the excision of a reduced vaginal cuff (median length 15 mm), is an independent prognostic factor, even if it did not affect the local recurrence rate [18]. These were the only two studies carried out with the same purpose as ours, but both of them have considered the FIGO stage I group (since the new classification in risk classes was much more recent), in which too many different variables are included together and could comprise all the four different ESMO-ESGO-ESTRO risk classes, leading to such different clinical management, according to the several prognostic factors included.

To our knowledge, this is the first study evaluating this prognostic factor in such a secluded and tidy cohort. According to our results, vaginal cuff length seems not to be an independent variable in EC low-risk group in terms of DFS, which disagrees with the previous study carried out in 2008, in which a more radical hysterectomy in stage I showed a lower rate of relapse and death [18]. However, our results reflect the ILIADE study cases, the only randomized trial addressing this issue, in which a more radical surgical approach, ensuring an excision of a wide vaginal cuff, did not reduce the risk of death or locoregional recurrence [4]. In the ILIADE study, although the length of the vaginal cuff excised in the Piver-Rutledge class II surgery arm was significantly longer than in the class I arm, the median length of vaginal cuff removed in the class I arm (15 mm) was considered probably sufficient by the authors to prevent vaginal recurrence [4]. This cut-off was pointed out by the retrospective study in 2008 [18], in which the excision of a small vaginal cuff (median length = 15 mm) was an independent prognostic factor in terms of disease-free and overall survival, although it did not affect the vaginal recurrence rate. Consistently, we referred to these previous studies [4,18] to choose vaginal cuff length cut-off. Unfortunately, probably due to the low numbers of recurrences in each group, we were not able to correlate vaginal cuff length with the site of recurrence: this potential correlation should be studied in a larger cohort of patients. It also has to be pointed out that the size of the vaginal neck can influence the quality of life (QoL) in patients undergoing a radical hysterectomy: several papers have evaluated the worsening of QoL after a radical hysterectomy [19,20,21]. Post-operative complications such as vaginal–bladder fistulas, rectal–vaginal fistulas, urinary tract dysfunctions and pain and numbness during sexual activity, may increase as the removed vaginal cuff length increases. Furthermore, it has been shown that sexual function decreased in patients undergoing a hysterectomy for EC treatment [20].

In detail, our results showed that vaginal cuff length does not influence the recurrence rate, which is probably determined by biomolecular factors [22], including L1CAM [23]. It has been described as critical in EC to promote the epithelial–mesenchymal transition and predictive of worse outcomes among EC, including tumors diagnosed at an early stage. L1CAM was described as the best ever published prognostic factor ready to greatly predict recurrence and death [24]. The incorporation of such molecular alterations (p53, Mismatch repair deficiency, and POLE, CTNNB1, L1CAM) into established clinicopathologic risk factors of EC resulted in a refined, improved risk assessment. Thus, the ESGO/ESTRO/ESP consensus in 2020 defined for the first time different prognostic risk groups integrating molecular markers [25]. In our cohort, 84.2% of all recurrences were L1CAM+. Furthermore, a trend was observed in terms of site of recurrence: L1CAM seems to be associated with a higher risk of distant recurrence. However, although these data support the negative prognostic role of L1CAM we had also previously demonstrated it to be responsible for a higher recurrence rate, especially at distant, in a subgroup of patients with a generally good prognosis [10]. However, we can find contrasting results regarding the measured levels of L1CAM gene/protein, for example, the contrasting results of proteomics and immunoblotting vs. gene expression [26,27]. In many studies, gene expression, proteomics, immunoblotting, and immunohistochemistry have been used for L1CAM, but results are not necessarily equivalent (for example, conclusions from gene expression are not necessarily equivalent to protein expression). This happens because design differences make it difficult to compare the findings of these studies, particularly when contrasting. Current biological knowledge accepts that genes/proteins involved in disease should be part of the same metabolic module. Indeed, genes and/or proteins driving the disease cooperate for its pathological progression. Biological functions and protein features should complement each other to improve metabolic performance [28,29].

A prospective randomized trial is needed to confirm its negative prognostic role, also in the EC low-risk group, with a potential value in clinical practice to detect that subgroup of low-risk patients which are at higher risk of recurrence, especially at distant.

However, larger studies are warranted, as PORTEC-4a (CT03469674), which will help evaluate the utility of treating patients based on molecular subtyping rather than clinicopathologic staging [25].

Its retrospective nature and the small number of recurrences given the rarity of low-risk endometrioid endometrial cancers that recur limit our study. However, our data are in line with the National Comprehensive Cancer Network (NCCN) guidelines, in which no recommendations in removing a vaginal cuff in FIGO stage I lesions are described [11], leading also to a better QoL in patients treated for EC.

## 5. Conclusions

Our data are in line with the National Comprehensive Cancer Network (NCCN) guidelines, in which no recommendations in removing a vaginal cuff in FIGO stage I lesions are described [11], leading also to a better QoL in patients treated for EC. Its retrospective nature and the small number of recurrences given the rarity of low-risk endometrioid endometrial cancers that recur limit our study.

## Figures and Tables

**Figure 1 cancers-14-00034-f001:**
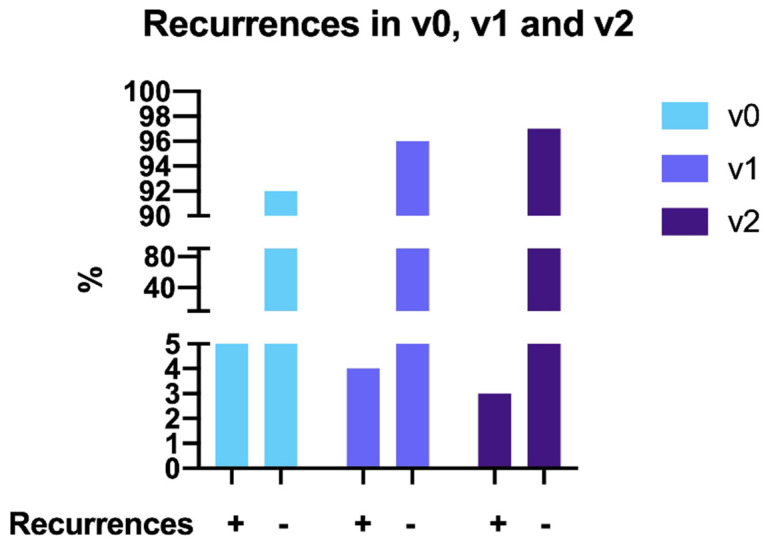
Recurrence sites in the three different groups: v0, v1, v2.

**Figure 2 cancers-14-00034-f002:**
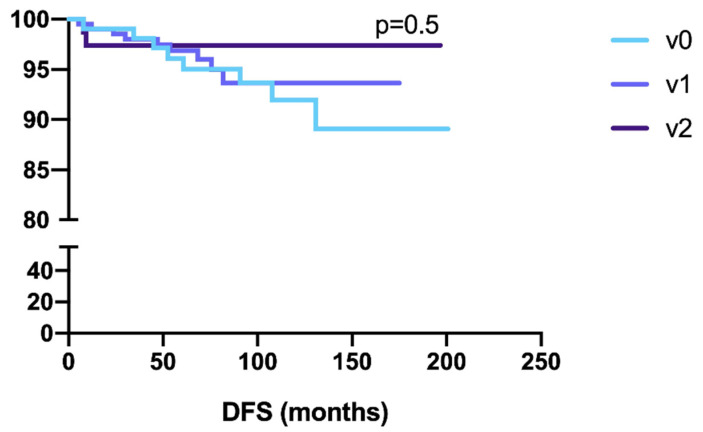
Ten years DFS in the three different groups: v0, v1, v2.

**Figure 3 cancers-14-00034-f003:**
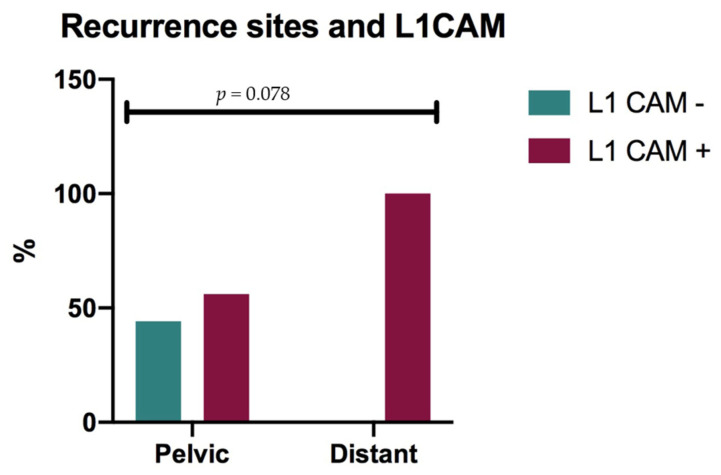
Recurrence site and L1CAM correlation.

**Table 1 cancers-14-00034-t001:** Patients’ clinical and pathological characteristics.

	V0	V1	V2	*p* Value
**N° patients**	100	179	65	/
**Median Age (years)**	58 (30–85)	60 (28–85)	58 (32–79)	n.s.
**Median BMI (Kg/m^2^)**	28.9 (17–51)	30 (16–53)	28 (18–51)	n.s.
**<30 kg/m^2^**	47 (47%)	87 (48.6%)	35 (53.8%)	/
** *≥* ** **30 kg/m^2^**	42 (42%)	86 (48%)	26 (40%)	
**No Value**	11 (11%)	6 (3.4%)	4 (6.2%)	
**Surgical approach**				*p* > 0.0001
**Laparotomy**	39 (39%)	25 (14%)	15 (23.2%)	
**Vaginal surgery**	3 (3%)	0	2 (3%)	
**MIS**	58 (58%)	154 (86%)	48 (73.8%)	
**Histologic subtype**				/
**Adenocarcinoma**	100 (100%)	179 (100%)	65 (100%)	
**Histologic tumor grading**				*p* < 0.01
**G1**	46 (46%)	68 (38%)	16 (24.6%)	
**G2**	54 (54%)	111 (62%)	49 (75.3%)	
**Median tumour size (mm)**	20 * (4–60)	20 (2–65)	20 * (7–70)	* *p* < 0.05

Results are presented as n (%) or median (range). BMI: body mass index. MIS: Minimally invasive surgery. * means the difference is between group V0 and V2.

**Table 2 cancers-14-00034-t002:** Recurrence characteristics.

Characteristics	Number (Range, %)
**N° patients**	21
**Median Age (years)**	68 (50–75)
**Media BMI (Kg/m^2^)**	27.4 (20–40)
**Surgical approach**	
Laparotomy	6 (28.6%)
Vaginal surgery	1 (4.8%)
Minimally invasive surgery	14 (66.6%)
**Histologic tumor grading**	
**G1**	6 (28.6%)
**G2**	15 (71.4%)
**Median tumour size (mm)**	20 (15–34)
**Recurrence site**	
Vaginal	7 (33.3%)
Lymph nodes	3 (14.3%)
Distant	11 (52.4%)
**Recurrence L1-CAM+**	
Vaginal	4 (57.1%)
Lymph nodes	1 (33.3%)
Distant	11 (100%)
No Value	2 (9.5%)
**Recurrence therapy**	
Surgery + RT	7 (33.3%)
Surgery + CT	3 (14.3%)
CT	11 (52.4%)
**Median DFS (months)**	47 (5.4–130.7)
**Median OS (months)**	74.6 (10.9–141.2)
**Follow-up Status**	
NED	5 (23.8%)
AWD	4 (19%)
DOD	12 (57.2%)

BMI: Body Mass Index; RT: Radiotherapy; CT: Chemotherapy; DFS: Disease free survival; OS: Overall survival; NED: No evidence of disease; AWD: Alive with disease; DOD; Death of disease.

**Table 3 cancers-14-00034-t003:** Recurrence sites in the three different groups: v0, v1, v2.

	V0	V1	V2	*p* Value
**Pelvic**	1/8 (12.5%)	5/10 (50%)	1/3 (33.3%)	NA
**Lymph nodes**	2/8 (25%)	1/10 (10%)	/	
**Distant**	5/8 (62.5%)	4/10 (40%)	2/3 (66.7%)	
**Total**	8/100 (8%)	10/179 (5.6%)	3/65 (4.6%)	ns (*p* = 0.28) *

Data are shown as n (%). * Chi-square test was performed for this comparison. NA: not applicable due to low number of recurrences. ns: not significant.

## Data Availability

The data presented in this study are available on request from the corresponding author. The data are not publicly available due to privacy.

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
