# Peer review of "Prognostic Role of the Removed Vaginal Cuff and Its Correlation with L1CAM in Low-Risk Endometrial Adenocarcinoma"

_cancers, 2021, doi:10.3390/cancers14010034_

Round 1

Reviewer 1 Report

The authors state that the aim of their study “was to investigate the role of the excised vaginal cuff length as a prognostic factor in terms of DFS and recurrence rate/site, in low-risk endometrial cancer (EC) patients.” In addition, they “correlated the recurrence with the expression of L1CAM.”  

They “used a retrospective data collection from March 2001 to November 2016 of women, undergoing surgical treatment for low-risk EC according to ESMO-ESGO-ESTRO Consensus.”

Authors conclude their results showed that the recurrence rate in patients with endometrial adenocarcinoma at low risk of recurrence and not receiving adjuvant therapy did not decrease with the increasing length of the removed vaginal cuff. The size of the removed vaginal cuff affects neither the site of recurrence nor survival. They also claim that their data is in line with the guidelines.

This result is not surprising because it is in line with something already known about the vaginal cuff and the EC. The vaginal cuff is the site of recurrent disease in Class I as well as Class II hysterectomy. Multiple authors have observed that vaginal cuff removal during abdominal hysterectomy appears as an independent prognostic factor in stage I endometrial carcinomas, nor does class II hysterectomy improve locoregional control and survival compared to class I hysterectomy. However, everyone agrees that a modified radical hysterectomy achieves optimal vaginal and pelvic disease control with a minimal increase in surgical morbidity. Indeed, extended surgery has been shown to reduce the incidence of vaginal recurrence of the disease.

The authors also hypothesize that the relapse rate is probably determined by biomolecular factors, including L1CAM. Starting from this basis, they used mouse monoclonal antibody anti-L1CAM to characterize the three sets of relapses and found a significant correlation with the presence of L1CAM (84.2% of all relapses). They also note that L1CAM appears to be associated with a higher risk of distant recurrence. However, larger studies are needed, they conclude, to endorse the usefulness of treating patients based on molecular subtyping rather than clinical-pathological staging.

The physio-pathological role of the vaginal cuff in endometrial cancer remains unclear because the molecular mechanisms underlying this cancer are scarcely clear, although some proteins (MSH2, MSH6, and L1CAM) are already used in clinics to diagnose patients with EC. However, even though it is an alternative approach to the clinic, there are some methodological uncertainties to be aware of.

The histological classification of tumor tissues does not always correspond to the “omics” or genomics classification of the same. Histology does not tell the where, when, and how a molecular function occurs in the examined tissue. Histology says that there are cancer cells in a tissue but does not explain the underlying molecular mechanisms. The heterogeneity of the sampling (absence of standards, difficulty in staging, and the proper sampling for controls) still makes it difficult to characterize the molecular profiles of cancer cells.

WE often find L1CAM involved in cancers and many studies aim at its molecular profiling, even in endometrial cancer. But in PubMed under the specific search “(L1CAM) AND (vaginal cuff)” I found no results. In endometrial cancer studies, many analyze gene datasets from TCGA and CPTAC, while others use immunohistochemistry staining or antibodies. Five genes (L1CAM, PRKCI, ESR1, CDKN2A, and VIM) are mostly studied and were included in a list to establish a formula for prognostic risk score.

However, about the measured levels of L1CAM gene/protein, we can find contrasting results, for example, the contrasting results of proteomics and immunoblotting vs gene expression (Lehrer vs Urick, 2021). In many studies, gene expression, proteomics, immunoblotting, and immunohistochemistry have been used for L1CAM, but results are not necessarily equivalent (for example, conclusions from gene expression are not necessarily equivalent to protein expression). This happens because design differences make it difficult to compare the findings of these studies, particularly when contrasting. Hence, a situation of uncertainty about the molecular mechanisms involving L1CAM in the EC is still poorly clear today.

When we go down to the molecular level to find the functional justifications of what we observe at the macroscopic-symptomatic level, we enter a different world, in the world of functional relationships, in the world of networks, where very different laws apply.

All this opens up to certain considerations; current biological knowledge accepts that genes/proteins involved in disease should be part of the same metabolic module. Indeed, genes and/or proteins driving the disease cooperate for its pathological progression. Biological functions and protein features should complement each other to improve metabolic performance (see Barabasi’s model). We can assess two certain proteins that will interact to originate a function through PPI networks. This is a central task in systems biology to achieve a holistic understanding of pathological processes. Network-level methods predict PPIs under the assumption that the network should appear connected, and nodes involved in disease should be part of the same network module. The logic (see always Barabasi’s model and its vast relevant literature) is that nodes of the same pathological module should be connected in a unique net around their hubs.

This logic also allows us to reverse-engineer networks when we have few genes or proteins involved in the same disease. We can use them as seeds to extract missing functional relationships from the entire genome. Enrichment analysis is useful for finding functional relationships in these subnets within genes linked to a single biological pathway where expression changes and functional relationships are additive. The rationale is that we should connect nodes of the same pathological module in a single network around their hubs.

L1CAM and the other proteins found involved in EC can be used through a reverse engineering process to recover the common network that underlies endometrial cancer. Ultimately, this is also a reliable check of the clinical hypotheses made at the macroscopic level of symptoms.

The authors can see this analysis in the attached pdf file. The results suggest that the functional data known to date doesn’t include L1CAM among the prognostic biomarkers of endometrial cancer.

Author Response

Dear referrer, first of all thank you for your review of the article, very careful and precise.

Major issues:

  • Regarding the vaginal length removed during hysterectomy, only the ILIADE study, in the literature, showed that vaginal length did not affect the loco-regional recurrence rate. Conversely, H. Arndt-Miercke at all. in 2008 demonstrated a correlation between DFS, OS and removed vaginal cuff. Our study is the only one that confirms the results of the prospective randomized ILIADE study. It also shows, with a median follow-up greater than 5 years, that the length of the removed vagina does not affect the site of recurrence. All of this is well reported in the discussion (page 9, lines 207-217)
  • Regarding the importance of L1CAM in endometrial cancers, integrated genomic characterization by the Cancer Genome Atlas (TCGA) in 2013 defined four distinct endometrial cancer subgroups (POLE mutated, microsatellite instability, low copy number, and high copy number) with possible prognostic value. The validation of surrogate markers (p53, Mismatch repair deficiency, and POLE) to determine these subgroups and the addition of other molecular prognosticators (CTNNB1, L1CAM) resulted in a practical and clinically useful molecular classification tool. The incorporation of such molecular alterations into established clinicopathologic risk factors resulted in a refined, improved risk assessment. Thus, the ESGO/ESTRO/ESP consensus in 2020 defined for the first time different prognostic risk groups integrating molecular markers. However, we have included your interesting observations and the references that you recommended to us in the discussion (page 10, lines 244-248 and pages 10-11, lines 253-262. References: 26-27-28-29)

Reviewer 2 Report

This article describes the clinical significance of length of vaginal cuff in surgical treatment for low-risk endometrial adenocarcinoma. This article provides a useful information about the surgical procedure for the patients with EC, although the number of patients with recurrence is small. There are several questions shown below.

  1. Line 201: “L1CAM was performed on all but two relapses and IT was positive in 71.4% of recurrences.” What does the abbreviation of IT mean?
  2. Line 215 and 217: “in which the median length of the excised vaginal cuff was mm 15 [4].” “median length mm 15”. Do authors mean 15 mm?

Author Response

  • Line 201: “L1CAM was performed on all but two relapses and IT was positive in 71.4% of recurrences.” What does the abbreviation of IT mean?We have corrected the spelling mistakes
  • Line 215 and 217: “in which the median length of the excised vaginal cuff was mm 15 [4].” “median length mm 15”. Do authors mean 15 mm?We have corrected the spelling mistakes

Round 2

Reviewer 1 Report

Certainly, the study of genes such as L1CAM requires a focused and in-depth study, outside this article. However, I believe that the interpretation that the authors give in the Discussion is correct and well-balanced. We must make controversial points known to the readers of the magazine who are not always experts in certain sectors. My opinion is now in favor of publication.